# Deep-Eutectic-Solvent-in-Water Pickering Emulsions Stabilized by Starch Nanoparticles

**DOI:** 10.3390/foods13142293

**Published:** 2024-07-21

**Authors:** Rongzhen Xie, Zhijian Tan, Wei Fan, Jingping Qin, Shiyin Guo, Hang Xiao, Zhonghai Tang

**Affiliations:** 1Hunan Engineering Technology Research Center for Rapeseed Oil Nutrition Health and Deep Development, College of Food Science and Technology, Hunan Agricultural University, Changsha 410128, China; rongzxie2024@163.com (R.X.); weifan@hunau.edu.cn (W.F.); gsy@hunau.edu.cn (S.G.); 2Institute of Bast Fiber Crops, Chinese Academy of Agricultural Sciences, Changsha 410205, China; tanzhijian@caas.cn; 3College of Bioscience and Biotechnology, Hunan Agricultural University, Changsha 410128, China; qinjingping@hunau.edu.cn; 4Department of Food Science, University of Massachusetts, Amherst, MA 01003, USA; hangxiao@umass.edu

**Keywords:** deep eutectic solvent -in-water, Pickering emulsion, starch nanoparticles, environmental friendliness

## Abstract

Deep eutectic solvents (DESs) have received extensive attention in green chemistry because of their ease of preparation, cost-effectiveness, and low toxicity. Pickering emulsions offer advantages such as long-term stability, low toxicity, and environmental friendliness. The oil phase in some Pickering emulsions is composed of solvents, and DESs can serve as a more effective alternative to these solvents. The combination of DESs and Pickering emulsions can improve the applications of green chemistry by reducing the use of harmful chemicals and enhancing sustainability. In this study, a Pickering emulsion consisting of a DES (menthol:octanoic acid = 1:1) in water was prepared and stabilized using starch nanoparticles (SNPs). The emulsion was thoroughly characterized using various techniques, including optical microscopy, transmission microscopy, laser particle size analysis, and rheological measurements. The results demonstrated that the DES-in-water Pickering emulsion stabilized by the SNPs had excellent stability and retained its structural integrity for more than 200 days at room temperature (20 °C). This prolonged stability has significant implications for many applications, particularly in the field of storage and transportation. This Pickering emulsion based on DESs and SNPs is sustainable and stable, and it has great potential to improve green chemistry practices in various fields.

## 1. Introduction

Pickering emulsions, distinguished by their stabilization through solid particles, represent a significant study area within colloid science. Traditional formulations of some emulsions predominantly use solvents as the organic phase and surfactants as the stabilizers. However, the use of these solvents often leads to environmental and health concerns owing to their potential toxicity and negative ecological impacts. Therefore, it is necessary to explore sustainable and safer alternatives [1].

In recent years, deep eutectic solvents (DESs) have been considered as greener substitutes for conventional solvents. They are typically composed of hydrogen bond donors (HBDs) and acceptors (HBAs) [2], which are mixed and heated at specific temperatures, and they are often considered to be a highly adaptable subclass of ionic liquids (ILs). Compared to traditional ILs, DESs offer advantages of easy preparation, low cost, and good environmental compatibility [3]. Furthermore, DESs can be derived from natural sources, which renders them less toxic than ILs [4]. Similarly to ILs, DESs can be categorized as either hydrophobic or hydrophilic, Additionally, DESs have the capability to effectively dissolve organic and inorganic substances [5,6]. Consequently, DESs have emerged as appealing alternatives to conventional solvents and ILs [7,8,9].

Recent advancements in the field of Pickering emulsions have successfully utilized DESs as oil phases. The use of DESs in water emulsions, particularly those stabilized with environmentally benign substances such as starch, is a significant step toward developing sustainable colloidal systems. The number of reported emulsions utilizing DESs as the oil phase, predominantly employing stable nonionic surfactants, is limited [10,11,12]. In this study, we present a novel DES-in-water Pickering emulsion formulation with advantages over previously reported formulations. Our approach eliminates the need for expensive and potentially toxic surfactants while maintaining low production costs. The only reported polysaccharide-stabilized DES-in-water emulsion was stabilized by cellulose nanofibrils [13]. Furthermore, as water is a major phase in our system, it is more suitable for applications requiring both low toxicity and cost-effectiveness compared to DESs in oil emulsions.

Through simple and cost-effective reaction steps, good compatibility is ensured while using non-toxic and easily available natural substances as solvent substrates and by adjusting specific proportions [4]. Menthol:octanoic acid (1:1) forms a hydrophobic DES with a density of 0.9 g/mL and a viscosity of 12.5 mPa·s [14,15]. It has the potential to extract and purify substances from water [16,17] and serve as a reaction medium for enzymatic esterification [18] for use in marine antifouling coatings [19].

Ball milling is a cost-effective and environmentally friendly technology that facilitates the modification of starch to nanoscale dimensions through mechanical actions such as shearing, friction, collision, and impact [20]. In comparison to other non-thermal processes, ball milling offers the distinct advantages of enabling fine particle grinding while maintaining low costs, simplicity in operation, enhanced safety measures, and reduced by-product discharge. Ball milling can change the particle shape and molecular weight of different starches [21].

The objectives of this study were to employ ball milling for starch modification, obtain nanoscale particles, and subsequently utilize these starch nanoparticles to stabilize emulsions containing menthol:octanoic acid DESs at a 10% volume ratio. Laser diffraction, visual observation, and rheological analysis were employed to assess the emulsions and their stability as well as to investigate the viability of this novel category of emulsions.

## 2. Materials and Methods

### 2.1. Materials and Reagents

Edible corn starch was provided by Hubei Yucheng E-Commerce Co., Ltd. (Huangshi, China) Menthol (purity: 99%) was obtained from Shanghai Aladdin Biochemical Technology Co., Ltd. (Shanghai, China), and octanoic acid (purity: 99%) was obtained from Shanghai Titan Scientific Co. Deionized water was used throughout the experimental procedures unless specified otherwise.

### 2.2. Preparation and Characterization of Starch Nanoparticle Suspension

Cornstarch was modified using a ball-milling method [22]. Appropriate amounts of cornstarch (20 g) and ultrapure water (400 g) were thoroughly mixed. Each sample of starch was poured into two 250 mL agate ball milling tanks. After 12 h of ball milling, the sample was poured into a cup to obtain a suspension of crude starch nanoparticles (SNPs). The sample was then ultrasonicated in an ultrasonic cell crusher (Ningbo Xinzhi Biotechnology Co., Ltd. (Ningbo, China)) at 50% power for 15 min, and the upper suspension was centrifuged at low speed to obtain an SNP suspension. For the characterization of SNPs, the particle size and distribution were precisely determined using a nanoparticle size analyzer. For this analysis, the samples were diluted extensively with ultrapure water, resulting in a 100-fold reduction in concentration. The measurements were conducted at a controlled temperature of 25 °C. The intensity of the scattered light was specifically observed at a 90° angle. Subsequently, the data obtained were meticulously analyzed using the software accompanying the instrument, enabling the calculation of both the average size and distribution range of the particle sizes. The morphology of SNPs was examined in detail using transmission electron microscopy (TEM). The samples had to be prepared into thin sheets or films. Subsequently, the samples were positioned on the pedestal of the transmission electron microscope, and the angle and position were adjusted to allow the electron beam to pass through the sample and project onto the detection screen by controlling the accelerator and collimator.

### 2.3. Preparation of DESs

In this study, DESs were prepared using menthol and octanoic acid in a 1:1 molar ratio as the HBA and HBD, respectively. The HBA and HBD were mixed in a round-bottomed flask and heated to 70 °C with continuous stirring for 1 h. A clear liquid was obtained after cooling. The density and viscosity of this DES were 0.9029 g/mL and 12.5 mPa·s, respectively.

### 2.4. Preparation of Pickering Emulsion

The prepared SNP suspension was diluted to five different concentrations using the gradient method. DESs were applied as oil phase, and SNP suspensions of different concentrations were mixed with the prepared DESs at various oil-to-water ratios. Subsequently, ultrasonic homogenization at 12,000 rpm for 1–2 min was performed using a homogenizer to form a white emulsion. To investigate the effect of particle concentration and oil-to-water ratio on the DES-in-water emulsion, a single factor test was conducted while keeping the other conditions constant.

### 2.5. Characterization of Pickering Emulsion

#### 2.5.1. Emulsion Fraction and Type

The emulsion fraction (defined as the volume ratio of cream to total volume) was recorded after being left undisturbed for 1, 7, 15, or 30 days. The emulsions were then carefully transferred to transparent sample bottles and hermetically sealed. Subsequently, they were stored for varying durations at ambient temperature to assess the emulsion fraction and evaluate their apparent storage stability. The emulsion type is determined by observing the behavior of a droplet when placed on the surface of water. If the droplet disperses in water, it indicates an oil-in-water (O/W) emulsion; otherwise, it suggests a water-in-oil (W/O) emulsion [23].

#### 2.5.2. Droplet Size and Microstructure

The DES-in-water emulsion was stored at ambient temperature, and the droplet size was determined by particle size analysis using a BT9300-HT laser particle size distribution instrument. For this analysis, prepare the emulsion sample to be tested and place it in the appropriate container. Then set the appropriate parameters for the instrument, such as refractive index, laser power, scattering angle, etc. The particle size distribution and average particle size of the emulsion droplets can be obtained through the instrument measurement and the analysis and processing of the obtained data. The observations and analyses were conducted using a NIKON Eclipse Ci optical microscope (Nikon, Japan).

#### 2.5.3. Rheological Measurement

The viscoelasticity of the DES-in-water emulsion was determined via strain and frequency scanning experiments using a super-rotating rheometer (Anton Paar, Graz, Austria). The rheometer was equipped with a 40 mm parallel plate with a geometric gap of 1 mm. For amplitude strain scanning, a fixed frequency of 1 Hz was employed while applying an oscillatory strain ranging from 0.01 to 1%. The apparent visco-shear rate curve was measured at a temperature of 25 °C and shear rates spanning from 0.01 to 1000 [1/s]. Additionally, the frequency scan encompassed a linear viscoelastic region with frequencies varying between 0.1 and 100 Hz, along with a constant strain level of 1%. All measurements were conducted at a temperature of 25 °C, and values for storage modulus (G′) and loss modulus (G″) were recorded [3,24].

### 2.6. Stability of Pickering Emulsion

To explore the stability characteristics of this novel starch-DES emulsion in aqueous environments, a series of comprehensive stability assessments were conducted. These evaluations are crucial for determining the robustness and practical applicability of emulsions under various conditions.

#### 2.6.1. Freeze–Thaw Stability

The emulsion underwent a rigorous freeze–thaw cycle, where it was initially subjected to a freezing temperature of −18 °C for 24 h. This phase was followed by a controlled defrosting process at ambient room temperature of approximately 20 °C for 2 h. This cycle was repeated twice to simulate real-world environmental fluctuations. The freeze–thaw stability of the emulsion was meticulously assessed through detailed visual inspection and precise droplet size analysis. This approach allows for a comprehensive understanding of both the dynamic and volumetric stability of emulsions under extreme conditions, as outlined in previous studies [25,26].

#### 2.6.2. Thermal Stability

To assess the thermal resilience of the emulsion, it was subjected to 100 °C for 10 min. During the thermal exposure, the emulsion was closely monitored for alterations in its appearance, microstructural integrity, and droplet size. These observations are pivotal for evaluating the thermal stability of the emulsion, offering insights into its behavior under high-temperature conditions, as referenced in previous studies [26].

#### 2.6.3. Centrifugal Stability

The stability of the Pickering emulsion under centrifugal force was evaluated systematically. The emulsion was placed in a 10-mL centrifuge tube and subjected to 2000 rpm at 15 °C for 5 min. The primary method for assessing the centrifugal stability was careful visual observation. This enabled determining the ability of the emulsion to maintain its structural integrity and appearance under the influence of centrifugal forces.

## 3. Results and Discussion

### 3.1. Characteristics of SNPs

The particle sizes of the SNPs were determined using a nanometer particle size analyzer. As depicted in Figure 1, the average particle size of SNPs was measured to be 192.13 nm. Previous studies have reported that SNPs prepared by the nanoprecipitation method exhibit sizes ranging from 50 to 300 nm, which aligns with our findings obtained using a nanoparticle particle size meter. During milling, starch undergoes constant stress and aggregation, leading to interactions under mechanical forces and various chemical reactions. The resulting SNPs generated via milling predominantly displayed irregular or tube bundle shapes and rough surfaces, with an observed particle size of approximately 200 ± 50 nm as confirmed by transmission electron microscopy (TEM) analysis (Figure 1b–d). Notably, a certain degree of aggregation was observed among the SNPs. Smaller particles contribute to enhanced stability when preparing a Pickering emulsion, while irregular shapes and rougher surfaces effectively hinder droplet aggregation and stabilize the Pickering emulsion [27]. The findings indicate that ball milling can produce nanometer-sized starch particles similarly to other chemical modifications, with the added benefit of a simpler and more convenient operation.

### 3.2. Pickering Emulsions Stabilized by SNPs

First, the stabilities of Pickering emulsions prepared with varying concentrations of SNPs were compared. Initially, the type of emulsion was determined by observing the dilution of the emulsion drops with water, indicating an O/W emulsion, in which the DESs served as the dispersed phase and water as the continuous phase. The accompanying Figure 2a,b display photos and provide results on emulsification. The appearance and fraction of different particle concentrations were recorded at 1-, 7-, 15-, and 30-day intervals. After a storage period of 7 days, stable SNPs exhibited distinctive variations in emulsion fraction and coverage, with the degree of emulsification dependent on the SNP particle concentration. According to Stokes’ law, the emulsification rate and the stability of an emulsion are directly proportional to the square of particle’s diameter. Consequently, as the droplet size decreased in the emulsion, the fraction of the emulsion increased. Moreover, a higher concentration of SNPs leads to a denser SNP layer at the interface, resulting in an enhanced emulsification rate and improved stability. This may be due to the fact that the viscosity of the water phase increases with the concentration of starch nanoparticles [28], making the emulsion become more stable [29]. Among the various oil/water ratios tested during storage, emulsions with 1:9 and 1:10 ratios exhibited superior stability. The SNP-stabilized DES-in-water Pickering emulsion showed excellent storage stability, which is consistent with the stability characteristics of general Pickering emulsions [23]. The emulsion exhibited long-term stability at room temperature and remained unchanged for over 200 days (Figure 2c). Wang et al. reported that a Pickering emulsion prepared with a 1:1 ratio and 3.0% starch concentration remained stable at room temperature for 30 days [30]. Moreover, emulsions with DES-to-water ratios of 1:9 and 1:10 demonstrated enhanced stability after storage for 30 days, with emulsification rates of 78.5 and 81.9%, respectively. Hadi et al. reported that modified quinoa starch with high chain length (propionylated and butyrylated), at higher modification levels, exhibited higher emulsification rates (>71%) and stability over the entire 50-day storage period. In contrast, our Pickering emulsion had excellent stability [31]. Consequently, we selected emulsions containing SNPs at a concentration of 2.7% and oil-to-water ratios of 1:9 and 1:10 for further analysis based on their superior performance during storage. The results underlined the critical influence of the particle concentration on the long-term stability of the emulsion and demonstrate the feasibility of starch-stabilized DES as a phase to form the emulsion.

Optical microscopy was used to observe the droplet size of the SNP-stabilized DES-in-water emulsions (Figure 3a,b). The droplet size observed using a microscope was approximately 15 μm. D(4,3) and D(3,2) are the basis for judging the droplet size, which are the average diameter of the volume and the average diameter of the surface area, respectively. For the emulsion with a ratio of 1:9, the D(4,3) and D(3,2) values were 15.5 and 10.35 μm, respectively, while for the emulsion with a ratio of 1:10, these values were 17.39 and 10.96 μm, respectively. Similar droplet sizes have been reported in previous studies on Pickering emulsions prepared using nanoparticles [32]. Yu et al. reported that a highly modified starch-stabilized Pickering emulsion remained stable after 30 days at 20 °C, and the droplet size ranged from 72.60 µm to 2.69 µm [33]. Marefati and Rayner reported that the droplet size of a Pickering emulsion stabilized by OSA-modified starch particles was 48.2 to 30.7 µm [34]. Compared to what they reported, our droplet size was smaller. The appropriate particle size may be one of the reasons for the long-term stability of the emulsion. These results demonstrate that SNPs effectively stabilize DESs as an oil phase in emulsions. The droplet size of the DES–water emulsion at a ratio of 1:10 was slightly larger than that with a ratio of 1:9, which may be attributed to properties such as hydrogen bonding within DES molecules [35]; thus, a smaller amount of DES leads to weaker molecular forces and, consequently, slightly larger particle sizes.

### 3.3. Rheological Analysis

To explore the applicability and fluidity of the emulsion, DES-in-water Pickering emulsions with oil–water ratios of 1:9 and 1:10 were selected for rheological analysis. The strain-scanning curves of the DES-in-water Pickering emulsion are shown in Figure 4a,b, prepared using varying proportions of DES and different concentrations of SNPs. The maximum value of G′ was within the range of 0.01–0.1%. Subsequently, we observed that G″ consistently surpassed G′, indicating a prevailing “liquid” behavior exhibited by SNP-based emulsions. Additionally, the pressure resistance of this type of emulsion is fragile. The dispersions with high particle concentrations exhibited higher G′ values than those with low particle concentrations after the intersection of G′ and G″ (yield stress point), indicating that the DES-in-water dispersions with high particle concentrations exhibited enhanced stability and stronger network formation under applied strain than those with low particle concentrations. With an increase in shear stress, G′ and G″ progressively diminished and eventually stabilized, potentially owing to localized or overall structural collapse within the sample [24,36]. This observation suggests that under intense stress, the properties of the emulsion become more delicate, which is possibly linked to the characteristics of the DES, such as its density and surface tension alterations, in response to pressure fluctuations.

Figure 4c,d show the apparent viscosity of the DES-in-water Pickering emulsions of different proportions of DESs to water and concentrations of SNPs. All emulsions showed the same obvious shear-thinning behavior as conventional emulsions [37], with the apparent viscosity decreasing as the shear rate increased. This phenomenon can be attributed to alterations in the microstructure of the emulsion during shearing. Compared to DES-in-water emulsions with lower particle concentrations, those with higher particle concentrations displayed elevated apparent viscosities at low shear rates, potentially due to enhanced network formation and improved stability of the emulsion droplets, which resulted from a higher particle concentration. At equivalent shear rates, emulsions containing a greater volume fraction of DESs demonstrated relatively higher apparent viscosity, possibly owing to the increased resistance associated with larger amounts of DES components. In the frequency scan (Figure 4e,f), G′ and G″ represent the viscoelasticity and emulsion strength, respectively, across the entire frequency range.

The values of G′ and G″ for the DES-in-water emulsions prepared with varying particle concentrations and oil-to-water ratios increased with increasing frequency. Specifically, at low frequencies (<1 Hz), the G′ value of the SNP-based DES-in-water emulsion surpassed that of G″, indicating a predominantly elastic behavior at this regime. This observation suggests that the oscillations occurring during this period did not disrupt the phase structure of the emulsion. When the frequency exceeded 1 Hz, the emulsion predominantly exhibited a viscous behavior. The DES-in-water emulsion with a higher particle concentration demonstrated an elevated G′ value, indicating improved elastic properties and a more stable network as particle concentration increased. Additionally, emulsions containing a higher proportion of the DES phase exhibited an increased storage modulus, suggesting that appropriate augmentation of the DES phase enhances elasticity. In summary, emulsions with a ratio of 1:9 displayed reduced droplet size and superior storage modulus performance. However, compared with the rheological properties of Pickering emulsions in previous reports, the DES-in-water Pickering emulsion has lower energy storage and strength, and is susceptible to deformation under the influence of external forces. It may be because DES as an oil phase is different from the oil phase in conventional Pickering emulsions. A conventional oil phase contains more complex compounds, but DES is formed by two single compounds through hydrogen bonding, resulting in a DES phase with a lower strength than that of a conventional oil phase. The advantage is that a DES-in-water Pickering emulsion may have greater potential in extraction and dissolution, and may also be more suitable for deformation or buffering materials.

### 3.4. Freeze–Thaw, Thermal, and Centrifugal Stability of Emulsions

The DES-in-water Pickering emulsion was packed into a 10 mL centrifuge tube at −18 °C, followed by defrosting at room temperature for 2 h after 24 h of freezing at −18 °C. The freeze–thaw stability of the emulsion was evaluated, and as depicted in Figure 5, negligible changes were observed in the appearance and droplet size compared to those before the freeze–thaw treatment. Specifically, the emulsion D(4,3) and D(3,2) of 1:9 were 17.56 and 10.91 μm, respectively, while those of 1:10 were 19.78 and 11.59 μm, respectively. The emulsion droplet size only slightly increased, indicating that SNP-stabilized DES-in-water emulsions exhibit excellent freeze–thaw stability. This could be attributed to the starch particles at the interface. The small starch nanoparticles at the interface were agglomerated to form larger starch particles. The relatively large starch particles provided a stronger and thicker barrier between droplets and protected their integrity during a freeze–thaw process [38]. Additionally, it could also be attributed to their unique properties, such as preventing aggregation of dispersed phases at low temperatures, coupled with the low melting point of the DESs, which prevented solidification under similar conditions.

The results shown in Figure 6a,b demonstrate the effects of the thermal treatment. At high temperatures, a noticeable breakage occurred in the emulsion, leading to stratification between the DES and water, with starch particles floating in the aqueous phase. The thermal stabilities of the SNP-stabilized DES-in-water emulsions were inadequate. This can be attributed to the structural alteration of starch at high temperatures, resulting in the disruption of molecular forces within the starch. Additionally, DES has superior properties in the extraction of polysaccharide, which may lead to instability in the emulsion prepared by polysaccharide particles under high temperature conditions [39,40].

After centrifugation, the emulsion exhibited notable changes (Figure 6c,d). Starch nanoparticles were effectively precipitated by centrifugation, resulting in complete demulsification of the emulsion. However, the centrifugal stability of the emulsion was found to be unsatisfactory.

The stability test results presented above demonstrate that the SNP-stabilized DES-in-water emulsions exhibit favorable storage stability at both low and normal temperatures. However, they are susceptible to destabilization under certain circumstances. Overall, the use of DESs in emulsion systems holds great promise for further research and less consumables, as the preparation method is simple.

## 4. Conclusions

This study introduces a novel approach that utilizes ball-milling techniques for altering the structure of starch at the nanoscale level. Starch nanoparticles were employed as stabilizing agents in a DES-in-water framework, culminating in the formation of a Pickering emulsion. This emulsion showed exceptional storage stability, preserving its composition at ambient temperature for up to 200 days. This SNP-based system outperformed existing emulsion systems in terms of stability over prolonged durations, a critically important feature for applications such as drug delivery systems, where controlled release is imperative. Moreover, it provides a more effective, eco-friendly, and sustainable method for emulsion formulation. Future research and development should concentrate on upscaling this technique for commercial use, refining the process to meet industrial demands and facilitating its widespread adoption in various sectors.

## Figures and Tables

**Figure 1 foods-13-02293-f001:**
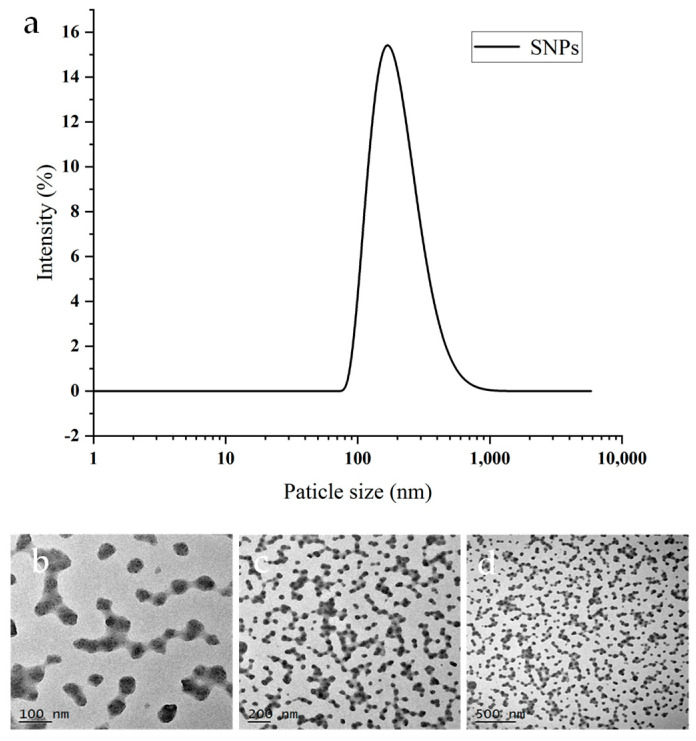
(**a**) Particle size and (**b**–**d**) TEM images of starch nanoparticles of different sizes.

**Figure 2 foods-13-02293-f002:**
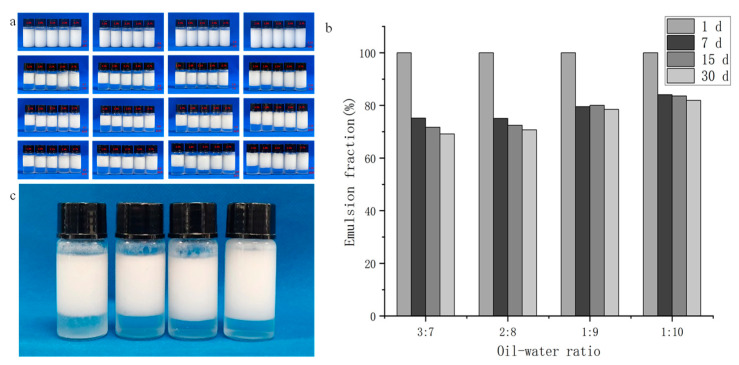
(**a**) Pictures of emulsions, from left to right, show the oil/water ratios of 3:7, 2:8, 1:9, and 1:10, and from top to bottom, show the storage for 1, 7, 15, and 30 days. (**b**) Emulsion fraction of a 2.7% SNP stable emulsion with different oil–water ratios. (**c**) DES-in-water emulsion stabilized by 2.7% SNPs beyond 200 days at room temperature.

**Figure 3 foods-13-02293-f003:**
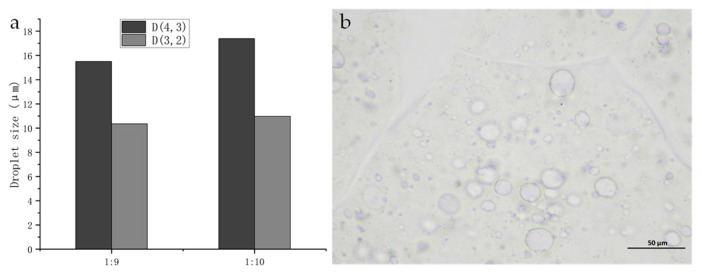
(**a**) Droplet size and (**b**) optical microscope image of emulsions.

**Figure 4 foods-13-02293-f004:**
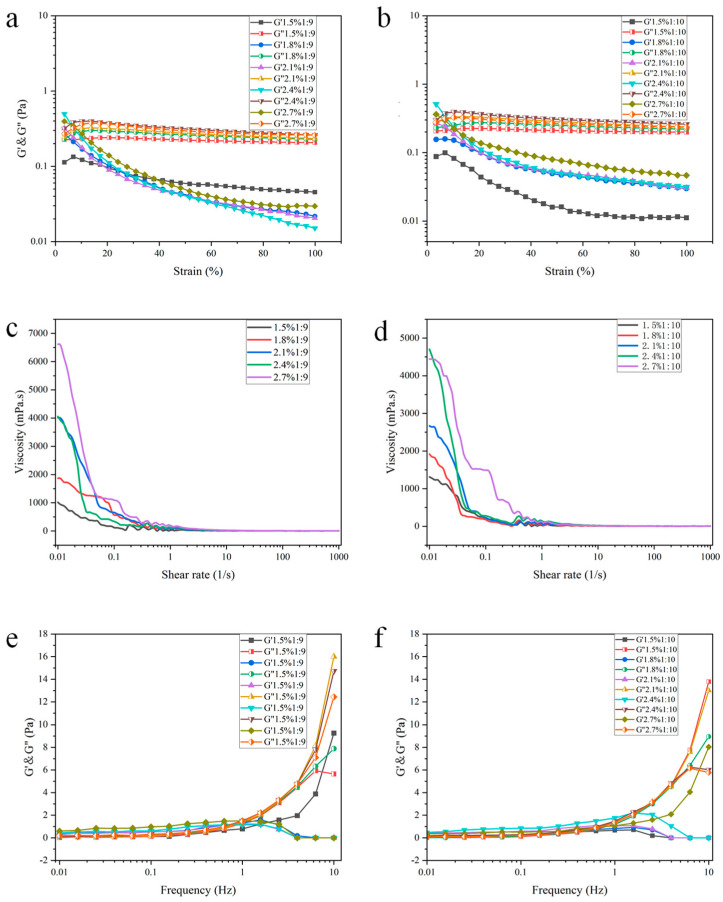
Rheological analysis diagram of emulsions. (**a**,**b**) is strain scanning of emulsion; (**c**,**d**) is the viscosity scan of the emulsion; (**e**,**f**) is the frequency scan of the emulsion.

**Figure 5 foods-13-02293-f005:**
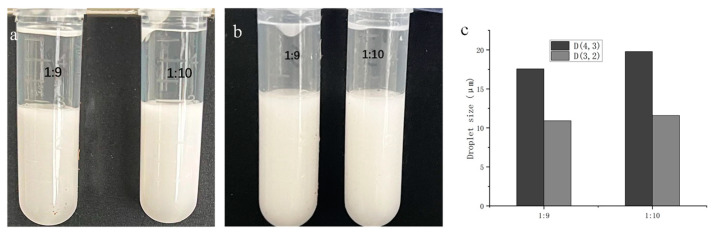
(**a**,**b**) Comparison of emulsion before and after freeze–thaw treatment, and (**c**) droplet size after freeze–thaw treatment.

**Figure 6 foods-13-02293-f006:**
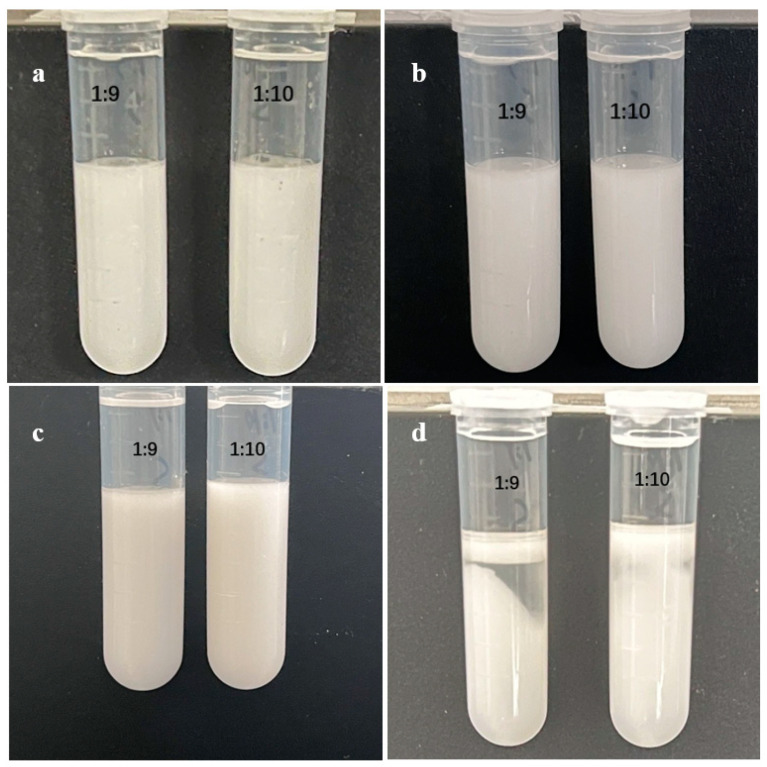
Emulsions after (**a**,**b**) thermal treatment and (**c**,**d**) centrifugation.

## Data Availability

The data presented in this study are available on request from the corresponding author due to privacy restrictions.

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
