# Peer review of "Deep-Eutectic-Solvent-in-Water Pickering Emulsions Stabilized by Starch Nanoparticles"

_foods, 2024, doi:10.3390/foods13142293_

Round 1

Reviewer 1 Report

Comments and Suggestions for Authors

Authors investigated the influence of deep eutectic solvents on preparation of pickering emulsions, using starch nanoparticles. The manuscript is interesting and it presents high novelty, however there are some missing points in manuscript that are required:

1. background in abstract should be re-written and put the higlight in better way why pickering emulsions are important, and what is the drawback so that they need advanced approach with deep eutectic solvents.

2. line 32: modify the sentence, because it is not correct. Authors said that oils and surfactants are toxic and that because of that they use deep eutectic solvents for emulsions. Oils are part of emulsions, and pickering emulsions are made actually to stabilize the oils, so they are not part of toxicity.

3. Methodology for Particle size distribution and tem for starch nanoparticle suspensions needs to be precisely described in experimental part.

4. Although the topic is interesting and authors obtained emulsions with high stability (over 200 days), they did not present any discussion in manuscript, just generic description of obtained results. Authors need to put discussion related to every charaterization technique to compare their results with conventional starch -based pickering emulsions, so to really demonstrate the benefit or draback of deep eutectic solvents in comparison to organic solvents/surfactants.

Reviewer 2 Report

Comments and Suggestions for Authors

Authors present research about Pickering emulsions based on hydrophobic DES and water, stabilized through starch nanoparticles obtained via ball milling. Generally, it is a work with impact of novelty, well planned and performed. No critical comments concerning to substantive part are claimed. 

However, there are some editorial or language remarks as follows.

1. The Abstract: a) change " as the stabilizers " onto " are the stabilizers " (l. 31); b) instead " menthol : octanoic acid = 1:9 " should be " menthol : octanoic acid 1:1 molar ratio " (l. 17) ; c) moreover, it would be valuable to add then information about DES/water ratio for which the Pickering emulsion exhibits optimal properties.

2. Instead " The oil phase was replaced with DES " (l. 104) better would be " As oil phase was applied DES ".

3. What authors mean writting " droplet polymerization between phases "  (l 177) ?

4. References list - many corrections or supplementations are necessary: a) l. 354, l. 371, l. 373, l. 380, l. 382, l. 401 and l. 427 - article nambers or pages are lacking; b) l. 362 - remove "American Chemical Society"  as its name is in a short form just before; c) l. 397 - change "0963-9969" onto relevant article number (i.e. 109870); d) l. 412 - change the high first word's characters onto low ones, consequently (moreover, remove "p." before pages numbers).

Comments on the Quality of English Language

See e.g. remarks mentioned at p. 1 and 2 of suggestions for authors.

Reviewer 3 Report

Comments and Suggestions for Authors

The paper refers to the characterization of o/w emulsion, where the oil phase was DES based on menthol and octanoic acid. Corn starch nanoparticles were used for emulsion stabilization. The topic is interesting, however, the manuscript requires improvement, e.g. important parameters in the description of the method are missing, and the results of additional experiments should be provided. The table collecting the compositions of prepared emulsions will be helpful.

Please find the detailed comments below.

Detailed comments:

- l. 42-43: “ to exhibit hydrophobic or hydrophilic properties, while effectively dissolving both organic and inorganic substances” suggests that organic substances are hydrophilic, and inorganic are hydrophobic – please rewrite

-l. 45-46: “incorporation of DES into oil phases” or apply DES as oil phases? Similarly in l. 104 – DES was applied as oil or “oil phase was replaced by DES”?

- l. 54-56 “Furthermore, given that water serves as a continuous phase in our system, it is more suitable than DESs in oil emulsions for applications requiring both low toxicity and cost-effectiveness.” – confusing, please rewrite this sentence

-l.58 – is menthol low-volatile substance?

- l. 63 – Ball – please correct.

- l. 79 – please provide the purity of octanoic acid

-l. 92 - rather “morphology” than “structure”

-l. 103 – as SNP do not dissolve in water, thus better” content” than “concentration”

-l. 108 “on the water-in-water DESs emulsion” – please correct

-l.109-116 – should be transferred to 2.5.1. point

- The details on droplet size, and particle size measurements should be provided

-l.176 - droplets cannot polymerize, please correct

-l 178 – ball milling is not “other chemical modification”

-l. 193-194 “Consequently, as the droplet size decreased in the emulsion, the fraction of the emulsion increased.” – data not provided

-l.202 “even at low SNP concentrations (as low as 1.5%).” – the data on used SNP content should be provided before this discussion

-fig. 2: presenting 1:2,3 and 1:4 instead of 3:7 and 2:8, respectively, would be more clear. Caption “oil-water” – please correct. Fig 2c- Why were four flasks presented for the same system?

-Fig. 3 – using the dye e.g. water phase contrast, would be beneficial

-l.218 “D(4,3) and D(3,2) values” – please explain

-l. 236-240 – Please provide the reference

-l.288-291 – This sentence is confusing: what are 17.56 and 10.91 micometers?

- Performing viscosity tests after freeze-thaw cycle is recommended. I

-l. 298 “a noticeable breakage occurred in the emulsion” – not visible in the figures

-l. 300-301- please explain

- is the system after heating up to 100°C still an emulsion? Please confirm.

-l. 302 “disruption of molecular forces within the starch” – please explain

-l.303 “extraction” – please explain

Comments on the Quality of English Language

Moderate editing of English language required

Round 2

Reviewer 3 Report

Comments and Suggestions for Authors

Comments on the Quality of English Language